# Effects of Reduced Extracellular Sodium Concentrations on Cisplatin Treatment in Human Tumor Cells: The Role of Autophagy

**DOI:** 10.3390/ijms25084377

**Published:** 2024-04-16

**Authors:** Laura Naldi, Benedetta Fibbi, Cecilia Anceschi, Patrizia Nardini, Daniele Guasti, Alessandro Peri, Giada Marroncini

**Affiliations:** 1Endocrinology, Department of Experimental and Clinical Biomedical Sciences “Mario Serio”, Careggi University Hospital, University of Florence, 50139 Florence, Italy; laura.naldi@unifi.it (L.N.); cecilia.anceschi@unifi.it (C.A.); alessandro.peri@unifi.it (A.P.); giada.marroncini@unifi.it (G.M.); 2Pituitary Diseases and Sodium Alterations Unit, Careggi University Hospital, 50139 Florence, Italy; 3Research Unit of Histology & Embryology, Department of Experimental & Clinical Medicine, University of Florence, 50134 Florence, Italy; patrizia.nardini@unifi.it (P.N.); daniele.guasti@unifi.it (D.G.)

**Keywords:** hyponatremia, human cancer cells, chemoresistance, DDP, autophagy

## Abstract

Hyponatremia is the prevalent electrolyte imbalance in cancer patients, and it is associated with a worse outcome. Notably, emerging clinical evidence suggests that hyponatremia adversely influences the response to anticancer treatments. Therefore, this study aims to investigate how reduced extracellular [Na^+^] affects the responsiveness of different cancer cell lines (from human colon adenocarcinoma, neuroblastoma, and small cell lung cancer) to cisplatin and the underlying potential mechanisms. Cisplatin dose–response curves revealed higher IC50 in low [Na^+^] than normal [Na^+^]. Accordingly, cisplatin treatment was less effective in counteracting the proliferation and migration of tumor cells when cultured in low [Na^+^], as demonstrated by colony formation and invasion assays. In addition, the expression analysis of proteins involved in autophagosome–lysosome formation and the visualization of lysosomal areas by electron microscopy revealed that one of the main mechanisms involved in chemoresistance to cisplatin is the promotion of autophagy. In conclusion, our data first demonstrate that the antitumoral effect of cisplatin is markedly reduced in low [Na^+^] and that autophagy is an important mechanism of drug escape. This study indicates the role of hyponatremia in cisplatin chemoresistance and reinforces the recommendation to correct this electrolyte alteration in cancer patients.

## 1. Introduction

Serum sodium (Na^+^) plays an important role in cell physiology. Together with potassium, Na^+^ is the major determinant of the regulation of cellular and extracellular volume. Serum Na^+^ alterations, particularly hyponatremia, are encountered in clinical practice in different scenarios [1,2].

Hyponatremia, defined by a serum sodium concentration ([Na^+^]) *<* 135 mEq/L, is the most frequent electrolyte imbalance in cancer patients, affecting up to 40% of them at admission to Oncology Units [3]. In this setting, the pathogenesis of hyponatremia is often multifactorial, with the syndrome of inappropriate antidiuresis (SIAD) representing the leading etiology [4]. The reduction in serum [Na^+^] secondary to SIAD is determined mainly by tumoral, ectopic secretion of arginine vasopressin (AVP) [5]. In addition, several classes of drugs, which are widely used in tumor patients, can induce AVP release. Among these, antidepressants, analgesics (especially opioids), anticancer treatments (chemotherapeutic agents, targeted agents, immune checkpoint inhibitors, and chimeric antigen receptor T cells), and palliative medications may contribute to reducing serum [Na^+^], possibly in association with underlying conditions (e.g., massive intravenous hydration during chemotherapy sessions, pain, and nausea) [5,6,7,8]. In the last decade, hyponatremia emerged as an indicator of a higher disease burden, a more compromised general status, with increased length of hospital stay and health costs [9,10,11,12], and a decreased progression-free and overall survival in patients with different malignancies [3,4,10,13,14,15,16]. On the other hand, normalization of low serum [Na^+^] is able to favorably affect clinical outcomes, prevent clinical complications, and reduce mortality [17], even in patients with extensive and terminal tumoral disease [4,18]. Therefore, hyponatremia represents a negative independent prognostic factor in the oncologic setting, and its use as a biomarker to identify high-risk patients affected by lung cancer has been proposed [19]. In agreement with clinical evidence, we have recently demonstrated for the first time that the reduction in extracellular [Na^+^] alters the homeostasis of different human cancer cell lines in vitro and in vivo, by promoting cell proliferation, invasion, and tumorigenicity [20,21,22,23,24].

Interestingly, beyond an effect on tumor initiation and progression, clinical evidence shows that cancer-related hyponatremia adversely affects the response to anticancer treatments, negatively conditioning the outcome of patients with different solid tumors as follows: metastatic renal cell carcinoma treated with everolimus [25], targeted therapy [26], or interleukin-2-based therapy [27], advanced lung cancer treated with first-line drugs [28,29] or immunotherapy [30], and hepatocellular carcinoma treated with sorafenib [31]. Interestingly, hyponatremia was demonstrated as an independent predictive factor of non-response to therapy (likewise to poor performance status and the absence of EGFR mutations in the tumoral tissue) in patients with non-small cell lung cancer (NSCLC) treated with the EGFR inhibitor erlotinib [32], thus strengthening the hypothesis of a direct impact of low serum [Na^+^] on resistance to pharmacological treatment. The therapeutic failure could be associated with the increase in cancer cell growth and aggressiveness induced by reduced extracellular [Na^+^] [20], but other factors might affect cell responsiveness to antitumoral drugs. To our knowledge, the pathophysiologic mechanisms that may underline the biological role of extracellular [Na^+^] on tumor cell sensitivity to pharmacological drugs have not been yet characterized.

Autophagy is one of the major mechanisms by which cancer cells resist chemotherapy [33]. It is a process of self-degradation by which cells can maintain homeostasis via the clearance of damaged or excess proteins and organelles in response to various stressors [34,35]. Autophagy tends to be activated in response to various chemotherapeutic treatments for solid tumors since it is closely associated with the metabolic adaptation pathways of tumor cells and the block of drug-induced apoptosis [36].

Among chemotherapeutic agents, platinum-based compounds (cisplatin, carboplatin, and oxaliplatin), which are used in the treatment of a variety of tumors, including lung cancer, colorectal, and pancreatic adenocarcinomas, and neuroblastomas [37,38], can both stimulate central AVP secretion and cause renal tubular injury and salt-wasting nephropathy [39], thus inducing hyponatremia or potentiating the reduction in serum [Na^+^] in previously hyponatremic subjects [40]. Hence, understanding the impact of hyponatremia on tumor cell resistance to platinum derivatives may be crucial to facilitate the process of treatment decision-making and warrant a more effective therapeutic plan.

The aim of the present study is to investigate the effects of reduced extracellular [Na^+^] on the response of different human cancer cell lines to cisplatin and the possible mechanisms that are involved.

## 2. Results

### 2.1. DDP and Cancer Cell Proliferation in Low Extracellular [Na^+^]

The cell lines used in this study are normally grown in a medium that contains 153 mM [Na^+^], which is then to be considered as a “normal” [Na^+^]. Cisplatin (cis-diamminedichloroplatinum, DDP) dose–response curves were performed in HCT-8, SK-N-AS, and H69 cell lines to identify IC50 variations in normal and low extracellular [Na^+^]. In particular, in HCT-8 cells, the IC50 for DDP at 153 mM [Na^+^] was 30.24 µM, and it increased upon lowering [Na^+^] with a peak of 64.15 µM at 115 mM (Figure 1a). Similarly, in SK-N-AS cells, starting from an IC50 of 10.75 µM in normal [Na^+^], an IC50 of 17.34 µM and 24.36 µM were reached at a [Na^+^] of 115 mM and 90 mM, respectively (Figure 1a). Likewise, in H69 cells, the basal IC50 for DDP was 28.16 µM, and it increased to 32.14 µM at 115 mM [Na^+^] and to 63.84 µM at 90 mM [Na^+^] (Figure 1a).

Colony formation assays were performed to further assess the ability of HCT-8 and SK-N-AS adherent cells to proliferate in the absence or presence of DDP. As previously demonstrated [18], the proliferation of HCT-8 and SK-N-AS cells increased in low [Na^+^] (115 mM) compared with 153 mM [Na^+^] (*p* ≤ 0.02 and *p* ≤ 0.002, respectively). As expected, DDP treatment at a normal [Na^+^] drastically reduced the number of cancer cell colonies in HCT-8 and SK-N-AS cells (*p* ≤ 0.05 and *p* ≤ 0.002, respectively). Conversely, DDP treatment at reduced [Na^+^] did not induce a statistically significant decrease in cell colonies in either cell line (*p* = 0.15 and *p* = 0.18, [Na^+^] 115 mM with DDP vs. [Na^+^] 115 mM without DDP in HCT-8 and SK-N-AS, respectively) (Figure 1b). Overall, these findings underline the lower efficiency of DDP treatment in low [Na^+^]. 

### 2.2. DDP and Cancer Cell Invasion in Low Extracellular [Na^+^]

Invasion assays were performed in order to evaluate whether DDP affects the invasiveness of cancer cell lines at different [Na^+^]. In order to emphasize the DDP effect, the number of invading cells was expressed as the ratio between DDP-treated and untreated cells normalized vs. the same ratio at 153 mM [Na^+^]. DDP treatment did not effectively counteract the invasiveness of all cancer cell lines when [Na^+^] decreased. The difference became statistically significant at 115 mM [Na^+^] in HCT-8 and SK-N-AS and was already present at 127 mM [Na^+^] in H69 cells (*p* ≤ 0.05 vs. 153 mM) (Figure 2a–c). These findings further indicate a reduced effect of DDP in low [Na^+^].

### 2.3. DDP and LC3 Expression in Low Extracellular [Na^+^]

Autophagy is a well-known mechanism of drug resistance in human cancer cells, and LC3 is one of the most important markers of autophagosome–lysosome formation. Western blot analysis was performed to assess the expression of LC3A/B isoforms I and II, which are structural proteins of autophagosome–lysosome membranes, widely used as biomarkers of autophagy. Particularly, the LC3 II/I ratio accurately represents the total autophagic flux since LC3-I, after cleavage by autophagin Atg7, produces the active membrane-bound form LC3-II.

In the absence of DDP, a trend toward an increased LC3 II/I ratio was observed in all cell lines upon lowering [Na^+^], with a statistically significant difference at 90 mM [Na^+^] (*p* ≤ 0.05 vs. 153 mM) (Figure 3a–c). The treatment with DDP amplified the difference in the LC3 II/I ratio between cells maintained in normal [Na^+^] vs. low [Na^+^], with a statistical significance starting from 127 mM [Na^+^] in all cell lines. Furthermore, when comparing the LC3 II/I ratio at the same [Na^+^] in the absence or presence of DDP, a trend toward higher values in the latter condition was observed. A statistically significant difference was observed at 90 mM [Na^+^] in HCT-8 cells and in all low [Na^+^] in H69 cells (Figure 3a–c).

Autophagosome–lysosome vesicles expressing cytosolic LC3A/B-I and membrane-bound LC3A/B-II were also analyzed by immunofluorescence at 153 and 115 mM [Na^+^] in HCT-8 and SK-N-AS cells receiving or not receiving DDP. The number of LC3A/B-positive pixels was significantly increased in low vs. normal [Na^+^] (Figure 4a,b). In addition, the immunofluorescent signals in cells cultured in low [Na^+^] (115 mM) and treated with DDP were more intense than in DDP-treated cells maintained in normal [Na^+^] (153 mM) (Figure 4a,b). Of note, when comparing the immunofluorescence at the same [Na^+^] in the absence or presence of DDP, in both cell lines, a statistically significant difference was observed in low [Na^+^].

### 2.4. Ultrastructural Analysis of Lysosomal Vesicles

Transmission electron microscopy (TEM) allowed us to visualize autophagosome–lysosome formation. This method revealed the presence of vesicles in the cytoplasm of HCT-8 (a), SK-N-AS (b), and H69 (c) cells. Larger lysosomal areas were observed in low [Na^+^] (115 mM) than in normal [Na^+^] (153 mM), with statistically significant differences in SK-N-AS and H69 cells (Figure 5a–c). In addition, the values in cells cultured in low [Na^+^] and treated with DDP were higher than in DDP-treated cells maintained in normal [Na^+^] (Figure 5a–c). Overall, the highest values were observed in cells cultured in low [Na^+^] and exposed to DDP.

### 2.5. Characterization of Autophagic Proteins

In order to better characterize the effect of DDP on the autophagic pathway, the expression levels of beclin 1, bcl2, and p62 were analyzed. Protein amounts were expressed as the ratio between DDP-treated and untreated cells normalized vs. the same ratio at 153 mM [Na^+^]. A trend toward increased expression levels of beclin 1, a key regulator protein of the autophagic process, was observed in reduced [Na^+^], with significant differences at 115 mM [Na^+^] in HCT-8 and SK-N-AS (Figure 6a–c). The expression of bcl2, a protein primarily involved in the regulation of apoptosis and that maintains autophagy levels within a physiological range, significantly increased at 115 mM [Na^+^] in all cells (Figure 6a–c). In HCT-8 cells, a significant effect was also observed at 127 mM [Na^+^]. p62 is a cargo receptor for ubiquitinated targets to autophagosomes for degradation; when its expression is upregulated, the degradation of autophagic lysosomes is impaired, resulting in a poor autophagic flux. Accordingly, p62 expression was significantly lower in low [Na^+^] (115 mM for HCT-8 cells and 90 mM for SK-N-AS and H69 cells) (Figure 6a–c).

## 3. Discussion

Hyponatremia is the most common electrolytic disorder in cancer patients and is widely considered a prognostic negative factor in tumor progression [4,41]. This can be, at least partially, due to clinical evidence showing that hyponatremia reduces the efficacy of chemotherapeutic intervention [25,26,27,28,29].

Platinum-based drugs are commonly used in anticancer strategies against different types of cancer [42]. In this study, we addressed the effect of low [Na^+^] on the response to cisplatin (DDP) in vitro. We used a previously validated experimental model of chronic hyponatremia [20] in order to maintain cells in low [Na^+^]. We performed the experiments in three different cell lines, i.e., HCT-8, SK-N-AS, and H69, which originated from tumors that were treated with platinum-based drugs, such as colon cancer, neuroblastoma, and small cell lung cancer, respectively [38,43,44].

First, we found that the half maximal inhibitory concentration (IC50) for DDP progressively increased in all cell lines upon lowering [Na^+^] compared with the IC50 observed in normal [Na^+^], thus indicating a reduced sensitivity to the drug. Noticeably, IC50 at least doubled in all cancer cells. The proliferation of HCT8 and SK-N-AS adherent cells was also assessed by colony formation assays. Accordingly, we demonstrated that DDP effectively reduced cell proliferation in normal [Na^+^], whereas it did not significantly affect cell growth in low [Na^+^].

We further analyzed the effect of DDP on HCT8, SK-N-AS, and H69 cell invasiveness at different [Na^+^]. As already demonstrated [4,20,22], the invasion ability of cancer cells markedly increased in low [Na^+^]. Noteworthily, the invasion assays also showed that the effect of DDP in counteracting cell invasiveness was blunted when cells were maintained in low [Na^+^].

Interestingly, we previously demonstrated that in low [Na^+^], an increased expression of the Na^+^–Ca^2+^ exchanger (NCX), together with a reversal of its activity, occurs [45]. In normal [Na^+^], the NCX operates by transporting three Na^+^ ions into the cell in exchange for one Ca^2+^ ion out of the cell. Instead, in low extracellular [Na^+^], the reversal of the NCX flux enhances Ca^2+^ accumulation into the cytoplasm. Autophagic dynamics in cancer are modulated by Ca^2+^ signaling [46].

Noteworthily, resistance to platinum-based drugs can be caused by different mechanisms that include autophagy [47]. There is in vitro evidence that autophagy is the main cause of resistance to DDP treatment in lung cancer [48,49]. Similar results have been also obtained in neuroblastoma cells [50,51] and colon cancer cells [43].

On the contrary, inhibition of autophagy is able to re-sensitize cisplatin-resistant tumor cells, thus enhancing the efficacy of chemotherapy [36]. This strategy has been proposed as an emerging therapeutic option to facilitate cisplatin sensitivity in neuroblastoma, colon rectal cancer, and SCLC [48,52,53].

Besides its role in inducing drug resistance in cancer, autophagy has been shown to play a role in cancer biology. Here, autophagy appears to play a controversial role because it can favor both cancer progression and inhibition. In particular, autophagy prevents tumor initiation and progression at the early stages of tumor development. However, at later stages, it induces tumor survival and growth and promotes invasion and metastasis formation [54]. Thus, cancer cells can exploit the benefits of autophagy by inducing cell survival associated with the stressful tumor microenvironment [55], as well as damage caused by chemo- and radiotherapies [56].

In view of the above-mentioned considerations, we hypothesized that low [Na^+^] might be associated with DDP resistance by affecting autophagic processes (Figure 7).

In order to assess the role of autophagy in mediating cisplatin resistance in cancer cells in low [Na^+^], we determined its effects on the expression of proteins that are involved in this process. Autophagy plays an important role in balancing sources of cellular energy, and LC3 (i.e., microtubule-associated protein 1A/1B-light chain 3) is commonly used as a marker for autophagosome–lysosome formation because of its dynamic behavior in this setting. There are two main forms of LC3 including the cytosolic form LC3-I and the membrane-bound LC3-II, which is often used as an indicator of autophagosome formation. Indeed, the conversion of LC3-I to LC3-II is a critical step in the progression of autophagy. Our data demonstrated that autophagy was triggered by low extracellular [Na^+^], as indicated by the significantly increased LC3A/B-II/I ratio at 90 mM [Na^+^] in all cell lines compared with normal [Na^+^]. Noteworthily, this difference was amplified after DDP treatment, thus supporting the role of autophagy in inducing cisplatin resistance in low [Na^+^]. These results were confirmed by the analysis of autophagosome–lysosome vesicles expressing the cytosolic LC3A/B-I and the membrane-bound LC3A/B-II isoforms via confocal immunofluorescence.

Noteworthily, the ultrastructural analysis of lysosomal areas further confirmed that autophagy was triggered by low [Na^+^], as it was enhanced upon DDP treatment in all the cancer cell lines used in this study.

Finally, we focused on additional key factors involved in the regulation of the autophagic process, namely, beclin 1, bcl2, and p62. Specifically, beclin 1 is involved in the first phase of phagophore formation [57]. Increased beclin 1 expression leads to increased autophagy and, in turn, to altered chemotherapy-induced apoptosis [58]. A major target of dysregulation in cancer cells that can occur during chemoresistance involves altered expression of b family members. Bcl-2 antiapoptotic molecules (bcl-2, bcl-xL, and mcl-1) are frequently upregulated in acquired chemo-resistant cancer cells, which block drug-induced apoptosis [59]. Indeed, decreased beclin 1 function is associated with the accumulation of the autophagy cargo binding protein p62 and altered NF-κB and inflammation signaling in tumorigenesis [60]. Thus, p62 accumulation is a result of autophagy inhibition [61].

In agreement with the above-described results, the LC3A/B downstream signaling pathways were modified upon DDP treatment. Notably, DDP induced a trend toward greater expression levels of beclin 1 that correlated with a trend in p62 reduction in low [Na^+^] in all cell lines. Similarly, the association between low sodium and DDP treatments promoted an increase in the anti-apoptotic bcl2 protein, with statistically significant differences at 115 mM [Na^+^] in all the tested cells. This finding is in agreement with our previous demonstration that apoptotic death is effectively inhibited in low [Na^+^] [22]. Overall, these results indicate that autophagy appears to be a determinant of DDP resistance in reduced [Na^+^].

In conclusion, our data demonstrate for the first time that low [Na^+^] induces chemoresistance to cisplatin in different cancer cell lines and that autophagy is an important mechanism of drug escape.

Admittedly, these findings may have clinical relevance, which is further reinforced by the knowledge that several chemotherapeutic drugs, including platinum-based agents, can induce hyponatremia. Therefore, a possible clinical scenario could be a situation in which a patient with cancer treated with a platinum-based drug develops hyponatremia, which in turn counteracts the antitumor effect of the drug itself. Overall, in view of the data presented in this study, it appears reasonable to strengthen the recommendation to monitor serum [Na^+^] regularly in cancer patients and to correct hyponatremia promptly when present.

## 4. Materials and Methods

### 4.1. Chemicals and Reagents

Small cell lung cancer (SCLC) (NCI-H69) human cell line (91091802), RPMI and Dulbecco’s modified Eagle culture media, fetal bovine serum (FBS), L-glutamine and antibiotics (penicillin and streptomycin), dimethyl sulfoxide (DMSO), WST-8 (2-(2-methoxy-4-nitrophenyl)-3-(4-nitrophenyl)-5-(2,4-disulfophenyl)-2H-tetrazolium, monosodium salt), and cisplatin (cis-diamminedichloroplatinum, DDP) (CAS 15663-27-1) were purchased from Sigma Aldrich SRL (Milan, Italy). Stromal (S)-type (SK-N-AS) neuroblastoma and colon adenocarcinoma (HCT-8) human cell lines were purchased from the American Type Culture Collection (Manassas, VA, USA).

### 4.2. Cell Cultures

SK-N-AS, HCT-8, and H69 stock cell lines were routinely cultured in DMEM or RPMI-1640 supplemented with 10% fetal bovine serum (FBS), L-glutamine and antibiotics (50 U/mL penicillin and 50 μg/mL streptomycin) and maintained at 37 °C in a humidified atmosphere (5% CO_2_/95% air). The choice of these cell lines was based on the fact that they had been already studied in previous studies focusing on low [Na^+^] and are representative of tumors responsive to platinum-based drugs, such as colon cancer, neuroblastoma, and SCLC, respectively [38,43,44]. To test the effects of low extracellular [Na^+^] on cancer cell lines, growth media with different [Na^+^] using a 2X DMEM sodium and glucose-free medium (PAA M-Medical, Milan, Italy) were prepared, as previously described [20,21,22,62,63,64]. The reference [Na^+^] for these cells was 153 mM, i.e., the concentration contained in DMEM. In the experimental protocol, extracellular [Na^+^] was progressively lowered by daily medium changes in order to adapt cells to [Na^+^] variations. Then, each experiment was performed upon culturing cells at the target [Na^+^] for 7 days before cisplatin treatment at a dose corresponding to the IC50 (30 μM for HCT-8, 11 μM for SK-N-AS, and 30 μM for H69 cells).

### 4.3. Analysis of Cell Proliferation and Viability

Dose–response curves were obtained by seeding 10^4^ cells/well in 96 well plates and treating with increasing doses of DDP (0 μM, 5 μM, 10 μM, 20 μM, 30 μM, 50 μM, 70 μM, and 100 μM) for 48 h. After 48 h of cisplatin treatment, the viability and proliferation of cells were assessed using Cell Counting Kit-8. Once metabolized, WST-8 creates a colored product (formazan), directly proportional to the number of metabolic and proliferative cells present in wells. The experiments were run according to the manufacturer’s protocol, and luminescence (490 nm) was recorded with a Wallac multiplate reader (Perkin-Elmer, Milan, Italy). The results were expressed as optical density (OD)/well (mean ± SE) and were normalized vs. 0 μM. The IC50 values were obtained using GraphPad Prism 5.0 (https://www.graphpad.com (accessed on 16 April 2021)), and triplicate analyses were performed at least three times.

### 4.4. Colony Formation Assay

Once the desired [Na^+^] was reached, each cell line was plated at a density of 10^3^ cells/well, and DDP was added at the concentration of the IC50 value (30 μM for HCT-8 and 11 μM for SK-N-AS). After 2 weeks, the colonies were washed with PBS three times, fixed with cold methanol, and stained with 0.5% crystal violet. The colony number was counted using an inverted microscope at 20× magnification.

### 4.5. Invasion Assay

Cancer cell invasive migration was assessed by a standard Transwell invasion system using polycarbonate filter inserts (8 μm pore size) (Corning^®^ Costar^®^ Transwell^®^ cell culture inserts, Corning, NY, USA), which were coated with BD Matrigel Basement Membrane Matrix (BD Becton, Dickinson and Company, Franklin Lakes, NJ, USA) 0.3% for 1 h at 37 °C. The upper chambers were seeded with human cancer cell lines (10^5^ cells/mL in 100 μL of serum-free media) for migration to the lower chamber with serum-added media, and the inserts were incubated at 37 °C overnight. After 24 h, the inserts were stained with crystal violet, and the invaded cells were observed under the microscope. Finally, the invaded cells on the lower side of the insert were decolored with an Extraction Buffer, and the dye mixture was measured by a spectrophotometer (Perkin-Elmer, Milan, Italy) at the Optical Density (OD) of 560 nm.

### 4.6. Western Blot Analysis

Cells were lysed in RIPA lysis buffer supplemented with a complete protease and phosphatase inhibitor cocktail, and the protein concentrations were determined using a Bradford protein assay. Cell lysates (20–40 μg of proteins) were fractionated by 12% Mini-PROTEAN TGX Stain-Free Precast Gels (Biorad, Hercules, CA, USA) and transferred onto a PVDF membrane (Immobilon, Billerica, Millipore, Burlington, MA, USA). After 1h of 5% milk blocking, the membranes were incubated with specific primary antibodies as follows: rabbit monoclonal anti-LC3AB (1:500, PA1-16931, Thermo Fisher Scientific, Waltham, MA, USA), mouse monoclonal anti-beclin 1 (1:1000, D40C5, Cell Signaling Technology, Danvers, MA, USA), mouse monoclonal anti-bcl2 (1:500, sc-7382, Santa Cruz Biotechnology, Dallas, TX, USA), and mouse monoclonal anti-SQSTM/p62 (1:1000, D5L7G, Cell Signaling Technology, Danvers, MA, USA). Primary antibodies were incubated overnight at +4 °C, and subsequently, the membranes were washed twice using PBS 1× and incubated with the specific secondary antibody conjugated to horseradish peroxidase (HRP-linked anti-mouse IgG, #7076 Cell Signaling or HRP-linked anti-rabbit IgG, #7074 Cell Signaling Technology, Danvers, MA, USA). Chemiluminescent images were acquired with a Bio-Rad ChemiDoc Imaging System (Biorad, Hercules, CA, USA), and proteins of interest were quantified and normalized versus stain-free acquisition. Densitometric analyses were performed using ImageJ- win64 Java 8 Software (https://fiji.sc (accessed on 5 August 2020)).

### 4.7. Immunofluorescence

Adherent cells were fixed in 10% neutral buffered formalin, permeabilized in 0.2% Triton-X-100, blocked in 5% normal goat serum and incubated with rabbit monoclonal anti-LC3A/B antibody (1:100, PA1-16931, Thermo Fisher Scientific, Waltham, MA, USA) and Hoescht 33342 (NucBlue Live cell Stain Ready Probes reagent, R37605, Invitrogen-Thermo Fisher Scientific, Waltham, MA, USA) overnight. The day after, sections were washed twice in PBS and stained with anti-Rabbit IgG Alexa Fluor488 secondary antibody for 1 h at room temperature in the dark. Finally, the slices were washed and mounted in fluorescent Prolong Gold Antifade medium (P36930, Invitrogen, Thermofisher Scientific, Waltham, MA, USA). Negative control sections (no exposure to the primary antibody) were processed concurrently with the other sections for all immunohistochemical studies. Images were acquired with a Leica Stellaris 5 confocal laser scanning microscope equipped with LAS X software 4.0.2 (Leica Microsystems, Wetzlar, Germany) using a Plan-Apo 63×/1.4NA oil immersion objective. Quantitative image analysis was performed by selecting randomly ~5 visual fields per slide using the same setting parameters (i.e., pinhole, laser power, and offset gain and detector amplification below pixel saturation). The numbers of LC3 were counted by using the open-source cell image analysis software ImageJ- win64 Java 8 (https://fiji.sc (accessed on 5 August 2020)) [41].

### 4.8. Transmission Electron Microscopy (TEM)

HCT-8 and SK-N-AS cells (5 × 10^5^) were pelleted by centrifugation at 600× *g* and washed twice in isotonic PBS. The pellets were fixed in Karnovsky’s fixative (cacodylate-buffered 4% glutaraldehyde and 2% paraformaldehyde (pH 7.4) and post-fixed with a solution of 1% OsO_4_ (Electron Microscopy Sciences, Hatfield, PA, USA). The samples were embedded in Epon 812 epoxy-resin (Sigma-Aldrich SRL, Milan, Italy) and cut with an RMC MT-X ultramicrotome (EMME3, Milan, Italy), collecting ultrathin sections (~700 nm thick) that were afterward stained with UranyLess (Electron Microscopy Sciences) and alkaline bismuth subnitrate solutions. The observation was performed with a JEM-1010 electron microscope (Jeol, Tokyo, Japan) at 80 Kv, and photomicrographs were acquired at 25,000× magnification using a MegaView III high-resolution digital camera equipped with an image analysis software Radius 2.2 (Emsis, Muenster, Germany). Morphometric analysis was expressed as the ratio between vesicles and total cell area (n = 5) and quantified using ImageJ software (https://fiji.sc (accessed on 5 August 2020)).

### 4.9. Statistical Analysis

Statistical analysis was performed by GraphPad Prism Software, Version 5.0 (https://www.graphpad.com (accessed on 16 April 2021)). The statistical significance of value differences was evaluated by two-way ANOVA followed by Bonferroni’s multiple comparison test. Student’s *t*-test was applied for the comparison of two classes of data as indicated in the figures. Data are reported as mean ± SE or as mean ± SEM, and a *p* ≤ 0.05 was considered significant.

## Figures and Tables

**Figure 1 ijms-25-04377-f001:**
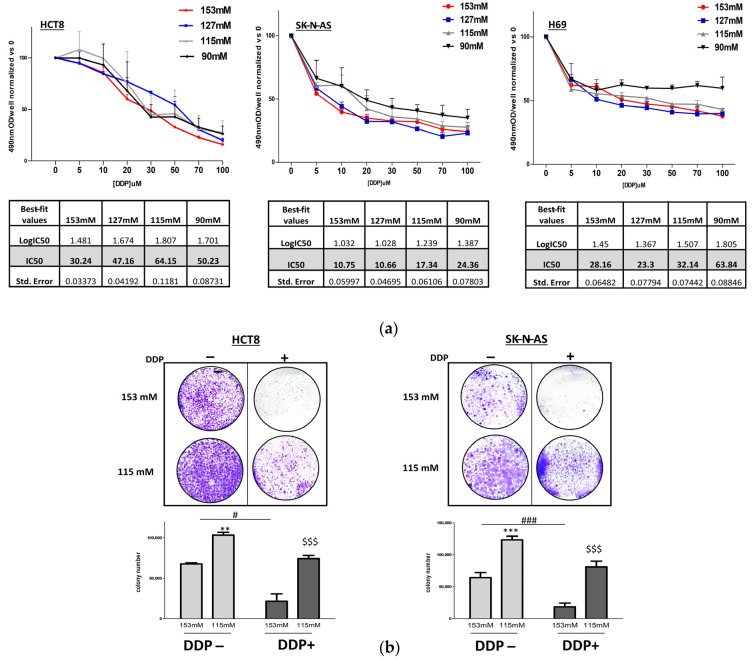
DDP effects on human cancer cell proliferation. (**a**) WST-8 dose–response curves at extracellular [Na^+^] of 153, 127, 115, and 90 mM after DDP treatment (0–100 µM). Graphs are plotted based on three independent experiments and tables summarize the IC50 values and standard errors resulting from all [Na^+^]. (**b**) Colony formation assays of HCT-8 and SK-N-AS adherent cells cultured at a [Na^+^] of 153 mM or 115 mM without (indicated with DDP−) or with DDP (indicated with DDP+) (30 µM and 11 µM, respectively, in the two cell lines). Microscopic images: representative examples from a single experiment for each cell line. Bars: colony numbers, expressed as mean ± SEM from three different experiments at 153 and 115 mM [Na^+^] (**, *** = *p* ≤ 0.02 and *p* ≤ 0.002 vs. 153 mM DDP−; #, ### = *p* ≤ 0.05 and *p* ≤ 0.002, 153 mM DDP+ vs. 153 mM DDP−; $$$ = *p* ≤ 0.002 vs. 153 mM DDP+).

**Figure 2 ijms-25-04377-f002:**
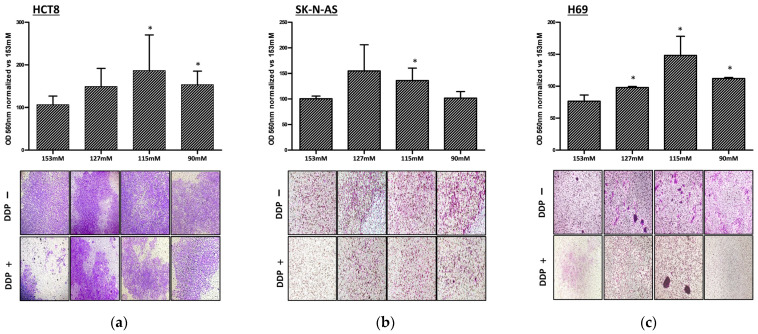
DDP effects on HCT-8 (**a**), SK-N-AS (**b**), and H69 (**c**) cell invasiveness. (**a**–**c**) The 560 nm OD/well of invading cells at [Na^+^] of 153, 127, 115, and 90 mM was expressed as the ratio between DDP treated/untreated cells, normalized vs. the same ratio at 153 mM [Na^+^]. Microscopic images from three representative experiments are reported in the lower panels (magnification 100×) (* = *p* ≤ 0.05 vs. 153 mM).

**Figure 3 ijms-25-04377-f003:**
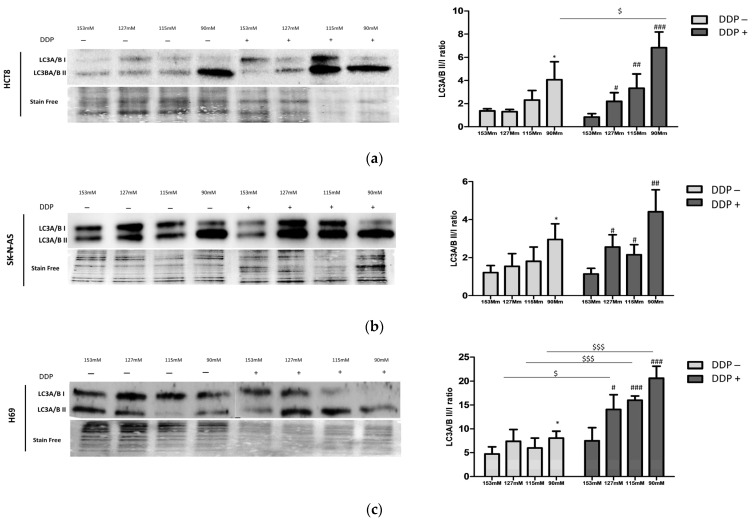
LC3A/B I-II protein expression. Western blot analysis of the LC3A/B II/I ratio at [Na^+^] of 153, 127, 115, and 90 mM in HCT-8 (**a**), SK-N-AS (**b**), and H69 (**c**). LC3A/B I and LC3A/B II protein expressions were normalized vs. stain-free blots. Data in the plots show the mean ± SEM (n = 3) (* = *p* ≤ 0.05 vs. 153 mM DDP−; #, ##, ### = *p* ≤ 0.05, *p* ≤ 0.02, and *p* ≤ 0.002 vs. 153 mM DDP+; $, $$$ = *p* ≤ 0.05 and *p* ≤ 0.002 DDP+ vs. DDP− for the same [Na^+^]).

**Figure 4 ijms-25-04377-f004:**
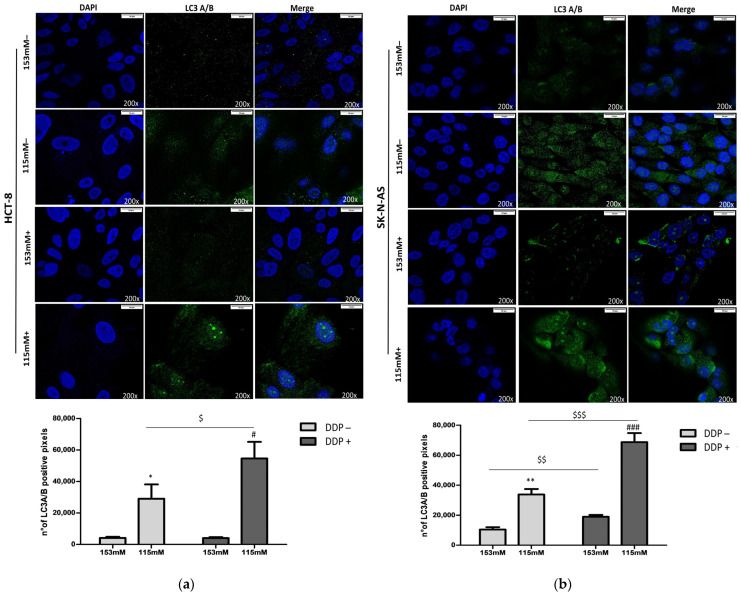
Confocal imaging of LC3A/B. Images showed the double stain of nuclei with DAPI (blue) and LC3A/B (green) at 153 and 115 mM [Na^+^] in HCT-8 (**a**) and SK-N-AS (**b**) cells. The LC3A/B positive pixels are quantified, and the bars show the mean ± SEM (n = 3) (*, ** = *p* ≤ 0.05 and *p* ≤ 0.02 vs. 153 mM DDP−; #, ### = *p* ≤ 0.05 and *p* ≤ 0.002 vs. 153 mM DDP+; $, $$, $$$ = *p* ≤ 0.05, *p* ≤ 0.02, and *p* ≤ 0.002 DDP+ vs. DDP− for the same [Na^+^]).

**Figure 5 ijms-25-04377-f005:**
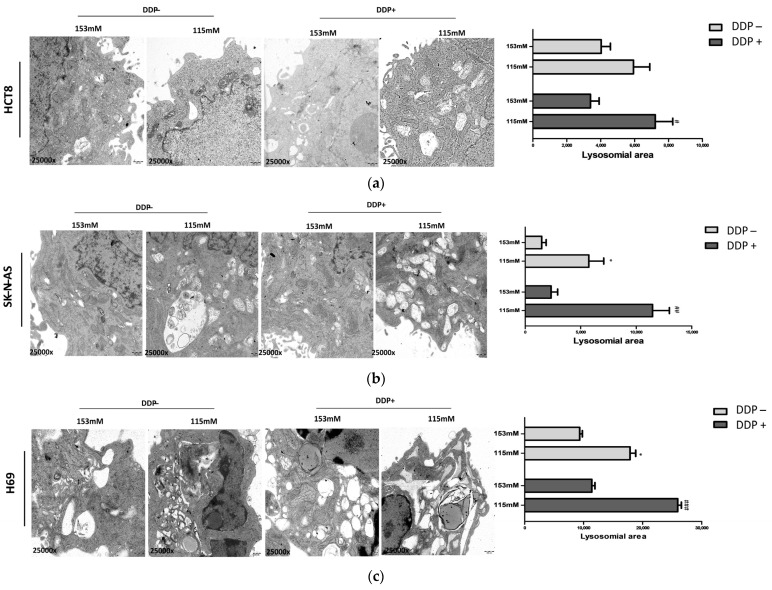
Transmission electron microscopy (TEM) analysis of autophagic vesicles. Images are indicative of cell samples grown at 153 mM or 115 mM [Na^+^] with or without DDP treatment (magnification 25,000×) in HCT-8 (**a**), SK-N-AS (**b**), and H69 (**c**) cells. Representative electron micrographs (**left**) and quantification (**right**) of lysosomal area per cell (n = 10 micrographs per each condition); one-way ANOVA followed by Bonferroni’s post hoc test, * = *p* ≤ 0.05 vs. 153 mM DDP−; #, ##, ### = *p* ≤ 0.05, *p* ≤ 0.02, and *p* ≤ 0.002 vs. 153 mM DDP+.

**Figure 6 ijms-25-04377-f006:**
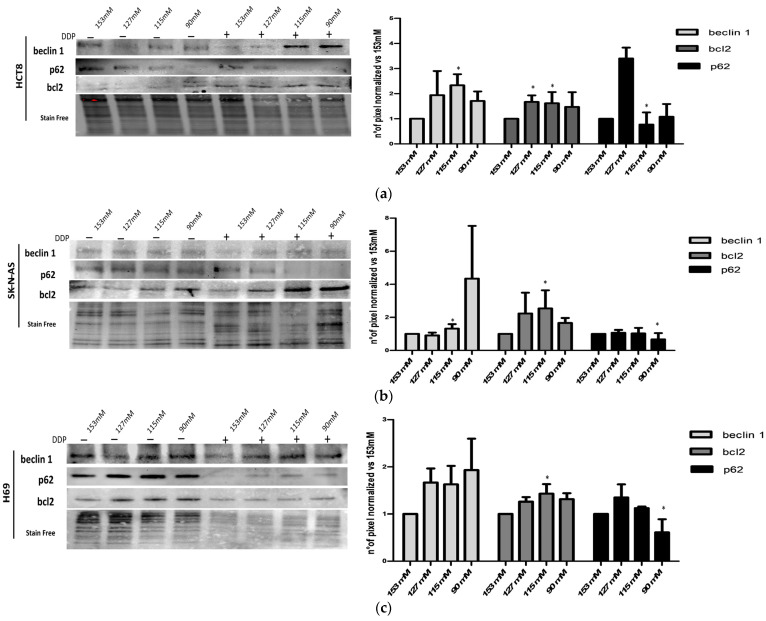
Western blot analysis of autophagic proteins (beclin 1, bcl2, and p62) in HCT-8 (**a**), SK-N-AS (**b**) and H69 (**c**) cells. Beclin 1, bcl2, and p62 protein expressions are normalized vs. stain-free blots. Images are representative of blots, and bars show the mean ± SEM of three different experiments (* = *p* ≤ 0.05 vs. 153 mM). Each [Na^+^] represents the ratio between DDP-treated and untreated cells.

**Figure 7 ijms-25-04377-f007:**
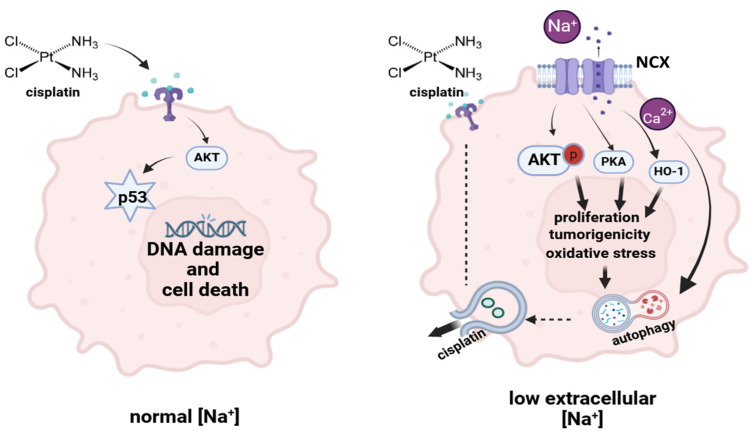
Schematic representation of the DDP mechanism of action in cancer cells in normal [Na^+^] (**left**) and the suggested mechanism of resistance in low [Na^+^] (**right**) (created with BioRender).

## Data Availability

The data and materials used to support the findings of this study are available from the corresponding authors (Benedetta Fibbi) upon request.

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
