# Peer review of "Effects of Reduced Extracellular Sodium Concentrations on Cisplatin Treatment in Human Tumor Cells: The Role of Autophagy"

_ijms, 2024, doi:10.3390/ijms25084377_

Round 1

Reviewer 1 Report

Comments and Suggestions for Authors

The aim of this manuscript is to investigate the effects of low sodium on the efficacy of cisplatin in several cancer cell lines, including CRC, GBM, and SCLC. The author tested different concentrations of sodium in culture media and found that lower than normal sodium concentrations in the culture media suppressed the effects of cisplatin on cancer cell proliferation and invasion. The potential mechanism was found to activate autophagy, which plays an important role in cancer cell drug resistance. This is an overall very interesting in vitro study.

  1. More clear labels should be provided on each figure, e.g., Fig. 1, sodium concentration, cisplatin should be labeled on the X, Y axes. This will aid the reader.
  2. High-quality images should be provided.
  3. What is the effect of low sodium on cell membrane potential?
  4. What is the effect of low sodium on Na/K-ATPase with or without cisplatin?
  5. A schematic figure presenting the potential mechanisms would benefit the readers.
  6. Is there any clinical evidence that hyponatremia may affect chemotherapy?
  7. The effect of sodium on cell physiology should be mentioned in the introduction.

Author Response

We thank the the reviewer for appreciating our study and for the comments.

With regard to the issues that have been raised:

  1. In each figure, labels have been modified as suggested by the reviewer.
  2. For figure 4, high-quality images were added.
  3. We have previously demonstrated a slight but not significant hyperpolarization in SK-N-AS cells cultured for 24 hours and 7 days in low sodium conditions (Hyponatraemia alters the biophysical properties of neuronal cells independently of osmolarity: a study on Ni(2+) -sensitive current involvement. Squecco R, Luciani P, Idrizaj E, Deledda C, Benvenuti S, Giuliani C, Fibbi B, Peri A, Francini F. Exp Physiol. 2016, 101:1086-1100).
  4. As previously demonstrated, the gene  expression  of  the Na+/K+-ATPase pump did not significantly change in SK-N-AS cells maintained in low sodium; for this reason, we did not analyse cisplatin effect on its regulation (Hyponatraemia alters the biophysical properties of neuronal cells independently of osmolarity: a study on Ni(2+) -sensitive current involvement. Squecco R, Luciani P, Idrizaj E, Deledda C, Benvenuti S, Giuliani C, Fibbi B, Peri A, Francini F. Exp Physiol. 2016, 101:1086-1100). Conversely, the above mentioned study underlined that low [Na+] induced an upregulation of the expression and the reversal of the activity of the Na+–Ca2+ exchanger (NCX).
  5.  We provided a schematic figure that summarizes the potential mechanisms of low sodium in cisplatin chemoresistance.
  6. As mentioned in the Introduction, cancer-related hyponatremia adversely affects the response to anticancer treatments and the outcome in different solid tumours, but to date the pathophysiologic mechanisms that may underline this biological effect have not been yet characterized.
  7. In the Introduction, we have mentioned the effect of sodium on cell physiology.

Reviewer 2 Report

Comments and Suggestions for Authors

The manuscript titled "Effects of reduced extracellular sodium concentrations on cisplatin treatment in human tumor cells: the role of autophagy" by Laura Naldi et al. explores the impact of low extracellular sodium (Na+) levels on the effectiveness of cisplatin (DDP) in treating various human cancer cell lines, with a particular focus on autophagy as a key mechanism behind the observed chemoresistance. Here are the comments.

While the introduction successfully establishes the context and significance of the research, it could benefit from a more detailed exploration of existing studies directly linking hyponatremia to chemoresistance mechanisms. This would provide a more precise rationale for focusing on autophagy as a critical mechanism of interest in the context of cisplatin treatment. The manuscript might have missed discussing recent literature that connects hyponatremia with cancer progression or treatment resistance beyond the scope of cisplatin and autophagy.

The method section could be enhanced by discussing the rationale behind selecting specific cell lines and the sodium concentrations used in the experiments. Additionally, justifying the choice of cisplatin concentrations for different assays would strengthen the methodological framework. 

While autophagy is well-studied, the manuscript could benefit from comparing or correlating autophagy with other stress responses or survival mechanisms in cancer cells under low Na+ conditions, such as apoptosis, necroptosis, or senescence.

Comments on the Quality of English Language

The overall quality of English is good

Author Response

We thank the reviewer for the comments and suggestions.

  • To date, there are not studies which directly link hyponatremia and molecular mechanisms of chemoresistance. In a previous study (Hyponatraemia alters the biophysical properties of neuronal cells independently of osmolarity: a study on Ni(2+) - sensitive current involvement. Squecco R, Luciani P, Idrizaj E, Deledda C, Benvenuti S,
    Giuliani C, Fibbi B, Peri A, Francini F. Exp Physiol. 2016, 101:1086-1100), we demonstrated that low [Na+] induced an upregulation of the expression and the reversal of the activity of the Na+–Ca2+ exchanger (NCX). Since the intracellular overload of Ca2+ correlates with autophagy, we hypothesized that low sodium could promote a Ca2+-mediated autophagic extrusion of cisplatin in cancer cells. Hence, we focused on the analysis of autophagy as a possible mechanism of cisplatin resistance in our model. This choice has been better elucidated in the text (Discussion).
  • For our experiments, we chose cell lines and [Na+] we had previously and extensively studied both in vitro and in vivo. For each cell line, experiments were performed using cisplatin IC50. These issues have been detailed in the Materials and Methods section.
  • We have previously demonstrated the effect of low [Na+] in affecting cell cycle and apoptosis in cancer cells (Marroncini G, Anceschi C, Naldi L, Fibbi B, Baldanzi F, Martinelli S, Polvani S, Maggi M, Peri A. Low sodium and tolvaptan have opposite effects in human small cell lung cancer cells. Mol Cell Endocrinol. 2021 Nov 1;537:111419). This has been mentioned in the Discussion.